# Immiscible inclusion bodies formed by polyglutamine and poly(glycine-alanine) are enriched with distinct proteomes but converge in proteins that are risk factors for disease and involved in protein degradation

Mona Radwan[1], Jordan D. Lilley[1], Ching-Seng Ang[2], Gavin E. Reid[1,3], Danny M. Hatters[1]*

**1** Department of Biochemistry and Molecular Biology; and Bio21 Molecular Science and Biotechnology Institute, The University of Melbourne, Parkville, VIC, Australia, **2** Bio21 Mass Spectrometry and Proteomics Facility, The University of Melbourne, Parkville, Victoria, Australia, **3** School of Chemistry, The University of Melbourne, Parkville, VIC, Australia

\* dhatters@unimelb.edu

## Abstract

Poly(glycine-alanine) (polyGA) is one of the polydipeptides expressed in Frontotemporal Dementia and/or Amyotrophic Lateral Sclerosis 1 caused by *C9ORF72* mutations and accumulates as inclusion bodies in the brain of patients. Superficially these inclusions are similar to those formed by polyglutamine (polyQ)-expanded Huntingtin exon 1 (Httex1) in Huntington's disease. Both have been reported to form an amyloid-like structure suggesting they might aggregate via similar mechanisms and therefore recruit the same repertoire of endogenous proteins. When co-expressed in the same cell, $polyGA_{101}$ and $Httex1(Q_{97})$ inclusions adopted immiscible phases suggesting different endogenous proteins would be enriched. Proteomic analyses identified 822 proteins in the inclusions. Only 7 were specific to polyGA and 4 specific to $Httex1(Q_{97})$. Quantitation demonstrated distinct enrichment patterns for the proteins not specific to each inclusion type (up to ~8-fold normalized to total mass). The proteasome, microtubules, TriC chaperones, and translational machinery were enriched in polyGA aggregates, whereas Dnaj chaperones, nuclear envelope and RNA splicing proteins were enriched in $Httex1(Q_{97})$ aggregates. Both structures revealed a collection of folding and degradation machinery including proteins in the $Httex1(Q_{97})$ aggregates that are risk factors for other neurodegenerative diseases involving protein aggregation when mutated, which suggests a convergence point in the pathomechanisms of these diseases.

## Introduction

The formation of protein inclusions is a hallmark of many neurodegenerative diseases. Inclusions are thought to derive primarily from the clustering of misfolded proteins into a

**Data Availability Statement:** All relevant data are within the manuscript and its Supporting Information files.

**Funding:** This work was funded by grants to DMH (National Health and Medical Research Council APP1161803 (https://www.nhmrc.gov.au/) and Motor Neuron Disease Research Institute, Australia small grant (https://www.mndaust.asn.au/Discover-our-research/About-MNDRIA.aspx)) and to DMH and GER (Australian Research Council DP170103093) (https://www.arc.gov.au/). MR acknowledges support from an Australian Government Research Training Program (RTP) Scholarship via the University of Melbourne (https://scholarships.unimelb.edu.au/awards/research-training-program-scholarship) and an Egyptian Ministry of Higher Education and Scientific Research PhD scholarship (http://portal.mohesr.gov.eg/en-us/Pages/default.aspx). The funders had no role in study design, data collection and analysis, decision to publish, or preparation of the manuscript.

**Competing interests:** The authors have declared that no competing interests exist.

centralized deposit [1]. While superficially many inclusions arising from different proteins appear similar in structure and morphology to each other, the mechanisms that mediate inclusion formation remains incompletely understood and involves multiple components. In the aggresome model, misfolded proteins are delivered by dynein-mediated transport to a centralized deposit near the microtubule organizing center [2]. However, other mechanisms must exist because other misfolded proteins form immiscible inclusion bodies when produced in the same cell [3–5]. Other models can explain multiple bodies as Q-bodies, JUNQ and iPOD structures, which have been suggested to operate as interconnected quality control processing centers for handling different classes of misfolded proteins (reviewed in [6]). One caveat with these models is that they explain inclusion assembly as primarily directed by cellular quality control mechanisms for proteins of different states of (mis) foldedness and less so by the physicochemical properties of the aggregating proteins. This is important in context of the physicochemical process of protein phase separation, which has emerged as a major mechanism to form membrane-less organelle-like condensates [7]. Phase separation into multiple immiscible phases may underlie, at least in part, the discrete inclusion structures seen by different misfolded proteins. Accordingly, different endogenous proteins may be directed to different aggregate phases based on either shared physicochemical properties or because they are recruited as part of quality control mechanisms to manage the formation or clearance of different aggregate phases.

Here we sought to assess whether two different and unrelated disease-associated mutant proteins that form superficially similar-appearing inclusions in cell culture, but which are immiscible in the same cell, share a similar or different pattern of co-recruitment of endogenous proteins. The two proteins include the exon 1 fragment of Huntingtin (Httex1), which accumulates into intraneuronal inclusions in Huntington Disease [8] and a dipeptide polymer of glycine-alanine (polyGA) that forms intraneuronal inclusions in Amyotrophic Lateral Sclerosis (ALS) and Frontotemporal Dementia (FTD) [9]. Both of these inclusions are SDS-insoluble and amyloid-like [10, 11].

The aggregation of mutant Httex1 is triggered by an abnormally expanded polyglutamine (polyQ) sequence encoded in exon 1 that arises by CAG trinucleotide repeat expansions [12, 13]. Long polyglutamine sequences form cytoplasmic or nuclear inclusions in animal and mouse models and are associated with a pathological cascade of events (reviewed in [1]). In FTD and ALS patients caused by *C9ORF72* GGGGCC hexanucleotide repeat expansion mutations, protein inclusions arise from the aggregation of polydipeptide repeats (PDRs) expressed abnormally from the expanded GGGGCC hexanucleotide repeat sequence. 5 different PDRs are expressed, namely dipeptide polymers of proline-arginine (polyPR), glycine-arginine (polyGR), proline-alanine (polyPA), proline-glycine (PolyPG) in addition to polyGA. Of these polyPR and polyGR are profoundly toxic when expressed in cell culture and animal models, with the toxicity targeting mechanisms in ribosome biogenesis, translation, and actin cytoskeleton among others [14–21]. PolyGA appears less toxic than the other PDRs although it has been reported to confer toxicity in some models [22–30]. PolyGA inclusions are however more widespread in FTD-ALS patient brain tissue compared to the other PDRs [9].

## Methods

### Plasmids

A pEGFP-based construct expressing polyGA dipeptide repeat length of 101 dipeptides (polyGA$_{101}$) was generated as described previously [21]. This construct expresses a GFP fusion tag at N-terminus of the polyGA. pT-REx vector expressing exon 1 of Htt (Httex1) with polyQ

sequence length of 97 and C-terminal mCherry or GFP fluorescent tags were prepared as previously described [31, 32].

## Cell lines

Neuro-2a cells, obtained originally from the American Type Culture Collection (ATCC), were maintained in Opti-MEM (Life Technologies). The medium was supplemented with 10% v/v fetal calf serum, 1 mM glutamine, and 100 Unit $mL^{-1}$ penicillin and 100 µg $mL^{-1}$ streptomycin, and cells were kept in a humidified incubator with 5% v/v atmospheric $CO_2$ at 37˚C.

## Transfections

Neuro2a cells were transiently transfected with the vectors using Lipofectamine 2000 reagent (Life Technologies). Specific transfection conditions for the different culture vessel types at densities of $9 \times 10^4$ (Ibidi 8-well µ-chamber) or $6 \times 10^6$ (T75 flasks). The following day cells (confluency of 80–90%) were transiently transfected with 1.25 or 60 µL Lipofectamine 2000 and 0.5 or 24 µg vector DNA, respectively, as per the manufacturer's instructions (Life Technologies). The next day, the medium was changed to Opti-MEM, and for the time course the medium was refreshed daily.

## Confocal imaging

Cells co-transfected with EGFPC2-GA$_{101}$ and Httex1Q$_{97}$-mCherry were fixed 24 h after transfection in 4% w/v paraformaldehyde for 15 min at room temperature. Nuclei were counterstained with Hoechst 33342 at 1:200 dilution (Thermo Fisher Scientific, San Jose, CA) for 30 min then washed twice in phosphate buffered saline (PBS). Fixed cells were imaged on a Leica SP5 confocal microscope using HCX PL APO CS 40× or 63× oil-immersion objective lens (NA 1.4) at room temperature. Laser used: 405 nm excitation, 445–500 nm emission–Hoechst 33342; 488 nm excitation, 520–570 nm emission–GFP; 561 nm excitation, 590 nm emission–mCherry. Single colour controls were used to establish and remove bleed through of the emission filter bandwidths. FIJI version of ImageJ [33] and Inkscape software were used for image processing.

## Purification of PolyGA and polyQ aggregates

Neuro-2a cells expressing either GFP-tagged GA$_{101}$ or Httex1Q$_{97}$ in 3 replicates were harvested by pelleting (200 $g$; 5 min; 24˚C) 24 h post transfection. Cell pellets were resuspended in lysis buffer (20 mM Tris, pH 8.0; 2 mM MgCl$_2$; 150 mM NaCl; 1% (w/v) Triton X-100; 20 Units/mL Benzonase, Novagen; 1× complete mini-protease cocktail; Roche) and then incubated for 30 min on ice. Lysates were diluted 2 times with PBS supplemented with protease inhibitor and aggregates were pelleted at 1000 $g$ for 6 minutes. The aggregates were washed twice with 1 mL PBS, then resuspended in 1 ml PBS and subjected to fluorescence-activated cell sorting (FACS) on a BD FACS Aria III instrument with an outlet nozzle of 100 µm in diameter. The flow rate was adjusted to ∼ 500 events/min, and EGFP fluorescence was monitored for sorting. Sorted aggregates were pelleted (12,000 $g$; 5 min; 4˚C), resuspended in PBS and washed 3 times by pelleting as above and resuspension in PBS. The final pellets were harvested by pelleting (21,000 $g$, 6 min, 4˚C) and dissolved in 10 µL neat formic acid for 30 min at 37˚C, vortexed for 20 seconds and sonicated for 1 min three times then incubated in a shaking microfuge tube incubator (30 min, 37˚C). Samples were neutralized to pH 7.0 by titration with unbuffered 3 M Tris. The protein concentration in the sample was determined by a Bradford assay using bovine serum albumin as mass standard. A total protein of 200 µg was further processed for mass spectrometry analysis.

## Collection of cells by pulse shape analysis

To assess the impact of polyGA aggregation on whole proteome, Neuro2a cells expressing GFP-tagged polyGA in 3 replicates were harvested 48 h post transfection by resuspension in PBS with a cell scraper. Cells were pelleted (120 $g$; 6 min) and resuspended in 2 mL PBS supplemented with 10 units/mL DNase I and filtered through 100 μm nylon mesh before analysis by flow cytometry. DAPI or Sytox (Thermo Fisher Scientific) was spiked into cell suspensions just before sorting to stain dead cells. Cells were analyzed by a FACS ARIA III cell sorter (BD Biosciences) equipped with 405-nm, 488-nm, 561-nm and 640-nm lasers. Live cells were gated using side and forward scatter as described previously [34]. Cells were further gated into cells with polyGA$_{101}$ in the soluble form (ni) and those with polyGA$_{101}$ inclusions (i) by pulse shape analysis (PulSA) as previously described [34]. Each gate recovered between $0.8–1 \times 10^6$ cells which were sorted directly into PBS and then snap frozen in liquid nitrogen and stored at–80˚C until used.

## Sample preparation for whole proteome analysis

Sorted cell populations were thawed and resuspended in 100 μl RIPA lysis buffer (25 mM Tris-HCl, pH 7.4, 150 mM NaCl, 1% v/v NP-40, 0.1% w/v SDS, 1% w/v sodium deoxycholate, 1× complete mini-protease mixture; Roche), and incubated on ice for 30 min. The concentration of proteins was measured by the Pierce microBCA Protein Assay according to the manufacturer's instruction (Thermo Fisher Scientific). Equal amounts of protein for each sample were precipitated with six volumes of pre-chilled (−20˚C) acetone, and incubation overnight at −20˚C. Samples were then pelleted (21,000 $g$, 10 min, 4˚C). Acetone was decanted without disturbing the protein pellet. The pellets were washed once with pre-chilled acetone then allowed to dry for 10 min. The protein precipitates were resuspended in 100 μl 0.1 M triethylammonium bicarbonate (TEAB) and were vortexed and then sonicated 3 times for 30 s. The samples were further processed for mass spectrometry analysis.

## Protein sample preparation for mass spectrometry

Proteins were subjected to reduction with 10 mM tris(2-carboxyethyl)phosphine (TCEP), pH 8.0, and alkylation with 55 mM iodoacetamide for 45 min, followed by trypsin digestion (0.25 μg, 37˚C, overnight). The resultant peptides were adjusted to contain 1% v/v formic acid then desalted by solid-phase extraction with an SPE cartridge (Oasis HLB 1 cc Vac Cartridge, Waters Corp., Milford, MA) pre-washed with 1 ml of 80% v/v acetonitrile (ACN) containing 0.1% v/v trifluoroacetic acid (TFA) and equilibrated with 1.2 ml of 0.1% v/v TFA three times. Samples were then loaded on the cartridge and washed with 1.5 ml of 0.1% v/v TFA before being eluted with 0.8 ml of 80% v/v ACN containing 0.1% v/v TFA and collected in 1.5 ml microcentrifuge tubes. Peptides were then lyophilized by freeze drying (Virtis, SP Scientific, Warminster, PA). The peptides were resuspended in 100 μl distilled water and quantified using microBCA assay with bovine serum albumin as the mass standard. Then, 10 μg of each sample (in a volume of 50 μl containing 100 mM TEAB) were differentially labelled by reductive dimethyl labelling using equal volumes (2 μl) of 4% light formaldehyde (CH$_2$O) or 4% medium formaldehyde (CD$_2$O, 98% D) and 0.6 M Sodium cyanoborohydride (NaCNBH$_3$). The peptide solutions were incubated on an Eppendorf Thermomixer (Eppendorf South Pacific Pty. Ltd., Macquarie Park, NSW, Australia) at room temperature for 1 h. After quenching with 8 μl of 1% v/v ammonium hydroxide followed by further quenching with 8 μl of neat formic acid, dimethyl-labelled peptides were combined in equal amounts prior to liquid chromatography-nano electrospray ionization-tandem mass spectrometry (LC-nESI-MS/MS) analysis.

## NanoESI-LC-MS/MS analysis

Peptides were analyzed by LC-nESI-MS/MS using an Orbitrap Lumos mass spectrometer (Thermo Fisher Scientific) fitted with nanoflow reversed-phase-HPLC (Ultimate 3000 RSLC, Dionex, Thermo Fisher Scientific). The nano-LC system was equipped with an Acclaim Pepmap nano-trap column (Dionex—C18, 100 Å, 75 μm × 2 cm) and an Acclaim Pepmap RSLC analytical column (Dionex—C18, 100 Å, 75 μm × 50 cm, Thermo Fisher Scientific). For each LC-MS/MS experiment, 1 μg (whole proteome) or 0.135 μg (aggregate proteome) of the peptide mix was loaded onto the enrichment (trap) column at a flow of 5 μl/min in 3% $CH_3CN$ containing 0.1% v/v formic acid for 6 min before the enrichment column was switched in-line with the analytical column. The eluents used for the LC were 5% DMSO/0.1% v/v formic acid (solvent A) and 100% $CH_3CN$/5% DMSO/0.1% formic acid v/v (solvent B). The gradient used was 3% v/v B to 20% B for 95 min, 20% B to 40% B in 10 min, 40% B to 80% B in 5 min and maintained at 80% B for the final 5 min before equilibration for 10 min at 3% B prior to the next analysis.

The mass spectrometer was operated in positive-ionization mode with spray voltage set at 1.9 kV and source temperature at 275˚C. Lockmass of 401.92272 from DMSO was used. The mass spectrometer was operated in the data-dependent acquisition mode, with MS spectra acquired by scanning from m/z 400–1500 at 120,000 resolution with an AGC target of 5e5. For MS/MS, the "top speed" acquisition method mode (3 s cycle time) on the most intense precursor was used whereby peptide ions with charge states ≥2 were isolated with an isolation window of 1.6 m/z and fragmented with high energy collision (HCD) mode, with a stepped collision energy of 30 ± 5%. Product ion spectra were acquired in the Orbitrap at 15,000 resolution. Dynamic exclusion was activated for 30s.

## Proteomic data analysis

Raw data were analyzed using Proteome Discoverer (version 2.3; Thermo Scientific) with the Mascot search engine (Matrix Science version 2.4.1). Database searches were conducted against the Swissprot *Mus musculus* database (version 2016_07; 16794 proteins) combined with common contaminant proteins. GFP sequence (UniProt ID: P42212) was also concatenated to the Httex1Q$_{97}$ and PolyGA$_{101}$ sequences. Search was conducted with 20 ppm MS tolerance and 0.2 Da MS/MS tolerance. The enzyme specificity was set as trypsin. The maximum number of missed cleavage sites permitted was two, and the minimum peptide length required was six. Variable modifications were used for all experiments: oxidation (M), acetylation (Protein N-term), dimethylation (K), dimethylation (N-Term), 2H (4) dimethylation: (K) and 2H (4) dimethylation (N-term). A fixed modification used for all experiments was carbamidomethyl (C). The false discovery rate (FDR) was calculated by the Percolator node in Proteome Discoverer v 2.3.0.81 and was set to 0.5% at the peptide identification level and 1% at the protein identification level. Proteins were filtered for those containing at least one unique peptide in all n = 3 biological replicates. Peptide quantitation was performed in Proteome Discoverer v.2.3 using the precursor ion quantifier node. Dimethyl labelled peptide pairs were established with a 2 ppm mass precision and a signal to noise threshold of 3. The retention time tolerance of isotope pattern multiplex was set to 0.6 min. Two single peak or missing channels were allowed for peptide identification. The protein abundance in each replicate was calculated by summation of the unique peptide abundances that were used for quantitation (light or medium derivatives). The peptide group abundance and protein abundance values were normalized to account for sample loading. In brief, the total peptide abundances for each sample was calculated and the maximum sum for all files was determined. The normalization factor was the factor of the sum of the sample and the maximum sum in all files.

After calculating the normalization factors, the Peptide and Protein Quantifier node normalized peptide group abundances and protein abundances by dividing abundances with the normalization factor over all samples. The normalized protein abundances were imported into Perseus software (v 1.6.5.0). Protein abundances were transformed to log2 scale. The samples were then grouped according to the replicates and protein intensities in these replicates were filtered, so at least two data points were present in total. For aggregate proteomes, proteins list was manually inspected to determine proteins that are completely specific to only one type of aggregate before missing quantitation values were filled with a constant (zero filling). For pairwise comparison of proteomes and determination of significant differences in protein abundances, paired Student's t test based on permutation-based FDR statistics was then applied (250 permutations; FDR = 0.05; S0 = 0.1). This was justified on the basis the proteomics abundance data was normally distributed.

## Bioinformatics

Protein interaction networks were generated using Cytoscape 3.7.1 [35] built-in STRING (v11.0) [36] using active interaction sources parameters on for Experiments, Databases, Co-expression neighborhood, Gene Fusion and Cooccurrence unless otherwise indicated. The minimum required interaction score setting was 0.9 (highest confidence) unless otherwise indicated. The corresponding enriched GO annotation terms were determined by calculating their enrichment *P*-value, which we compute using a Hypergeometric test, as explained in [37]. The *P*-values are corrected for multiple testing using the method of Benjamini and Hochberg [38]. Selected GO terms were used to manually re-arrange nodes and were added to protein interaction network using Inkscape.

IUPred [39] were applied to predict the intrinsically unstructured/disordered regions of proteins significantly enriched in polyGA$_{101}$ or Httex1Q$_{97}$ aggregates. Glutamine content was analyzed with the web-server COPid [40] (http://crdd.osdd.net/raghava/copid/whole_comp.html). A control set of 100 random proteins (S1 Table) was generated from a list of the mouse proteome obtained from the Uni-ProtKB database (http://www.uniprot.org/uniprot/?query=reviewed:yes+AND+organism:10090&random=yes). The Mann-Whitney- Wilcoxon test was employed to determine significant differences.

## Statistical analysis

The details of the tests were reported in the figure legends. All statistical analyses were performed with GraphPad Prism v 7.05 (Graphpad Software Inc., San Diego, CA). Significant results were defined on the figures for $P < 0.05$.

## Data availability

The MS proteomic data have been deposited to the ProteomeXchange Consortium via the PRIDE [41] partner repository with the dataset identifiers PXD018505 for aggregate proteome data and PXD018824 for whole proteome data.

## Results & discussion

We previously reported polyGA$_{101}$ to be mildly toxic to cultured Neuro2a cells and to induce a distinct network of proteome changes that occur compared to the arg-rich PDRs [21]. We also noted a distinction to the other PDRs in forming large inclusions that are morphologically similar to the inclusions formed by polyQ. When we co-expressed Httex1Q$_{97}$ as a fusion to mCherry, we found the polyGA$_{101}$ and Httex1Q$_{97}$ formed discrete inclusions in the same cell

with no apparent colocalization (Fig 1A–1B). This suggested that any concomitant co-aggregation patterns that arise with endogenous proteins may involve highly distinct proteins even though both proteins seem to form amyloid-like fibrils.

To investigate these potential differences, pellets recovered from lysates of neuro2a cells expressing GFP-tagged Httex1Q$_{97}$ or GFP-tagged polyGA$_{101}$ were sorted to purify the aggregates using flow cytometry via monitoring the GFP fluorescence. Quantitative proteomics, by way of dimethyl isotope labelling, was used to define the proteins enriched in each aggregate class (Httex1Q$_{97}$ versus polyGA$_{101}$) after normalization to total mass of protein. We observed 822 proteins in both inclusions (S2 Table). Of these 70 were significantly enriched in polyGA inclusions (3 replicates, a permutation-based FDR cut-off of 5% and S0 of 0.1) and 51 were enriched in Httex1Q$_{97}$ (Tables 1 and S2 and Fig 2A). 7 proteins were identified exclusively in polyGA inclusions (Phlda1, Etfa, Cbx4, Soga3, Vasp, Cops7a and Dtx3l) and 4 proteins identified exclusively in Httex1Q$_{97}$ inclusions (Pcnt, Lsm12, Specc1l and Arfgap2). We also noted the GFP moiety (which was identified from both Httex1 and polyGA fusions) was enriched in the polyGA aggregates suggesting that the inclusions formed by polyGA contain less mass of associated proteins than those formed by Httex1. This could arise by more non-specific interactions to the Httex1 inclusions or by polyGA forming a greater proportion (by mass) of the inclusion composition than for Httex1 inclusions.

Of the proteins found uniquely in polyGA or polyQ inclusions, several of the proteins appeared unconnected to other proteins in the protein-protein interaction analysis (Fig 2B).

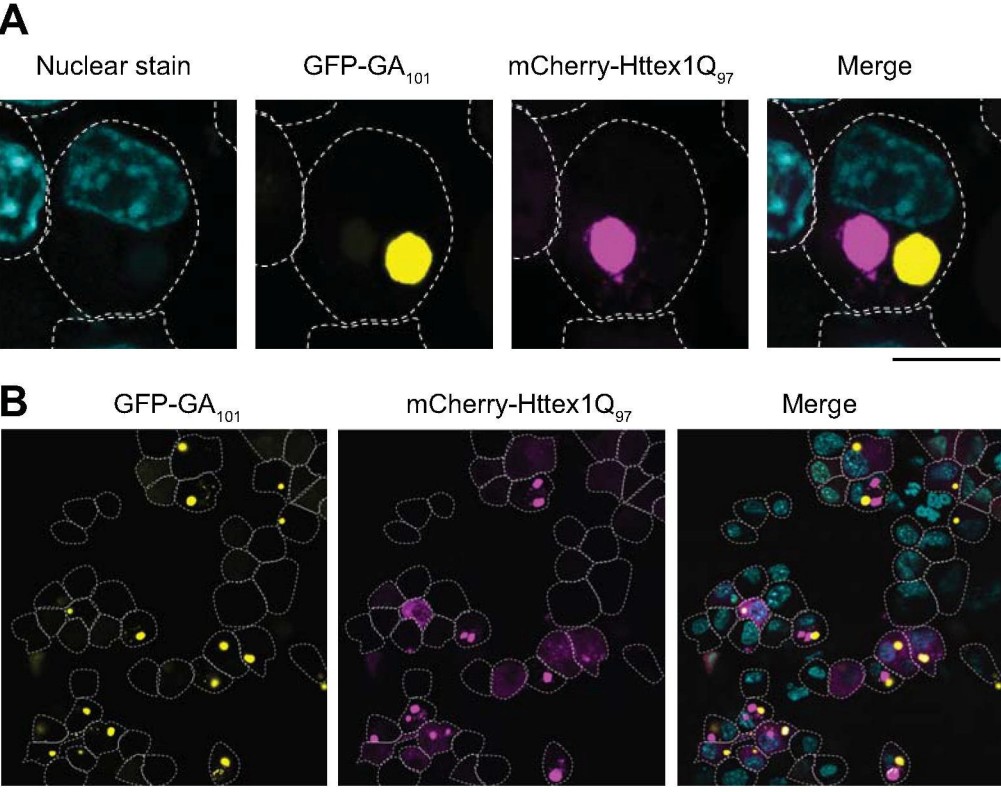

**Fig 1. Httex1Q$_{97}$ and polyGA$_{101}$ form distinct inclusions in neuro2a cells.** Confocal micrographs of neuro2a cells co-expressing GFP-tagged GA$_{101}$ (yellow) and mCherry-tagged Httex1Q$_{97}$ (magenta), fixed 24 hr post-transfection and stained with Hoechst33258 (cyan) to visualize nuclei. The dotted white lines show the outlines of cells (manually traced). Scale bar represents 5 μm. Panels **A** and **B** show two different scales of view.

**Table 1. Proteins enriched in inclusions of polyGA101 and Httex1Q97[*].**

| Enriched in polyGA$_{101}$ | | | Enriched in Httex1Q$_{97}$ | | |
|---|---|---|---|---|---|
| *Description* | *Gene ID* | *log2 enrichment (mean ± SD)* | *Description* | *Gene ID* | *log2 enrichment (mean ± SD)* |
| **Proteins uniquely identified in either of the aggregate type** | | | | | |
| Pleckstrin homology-like domain family A member 1 | Phlda1 | 8.91±0.11 | Pericentrin | Pcnt | 5.33±0.84 |
| Electron transfer flavoprotein subunit alpha, mitochondrial | Etfa | 7.24±0.70 | Protein LSM12 homolog | Lsm12 | 5.90±0.24 |
| E3 SUMO-protein ligase CBX4 | Cbx4 | 7.21±1.16 | Cytospin-A | Specc1l | 7.04±0.42 |
| Protein SOGA3 | Soga3 | 6.96±0.10 | ADP-ribosylation factor GTPase-activating protein 2 | Arfgap2 | 8.33±0.29 |
| Vasodilator-stimulated phosphoprotein | Vasp | 6.88±1.15 | | | |
| COP9 signalosome complex subunit 7a | Cops7a | 6.83±0.42 | | | |
| E3 ubiquitin-protein ligase DTX3L | Dtx3l | 5.87±1.67 | | | |
| **Proteins relatively enriched in either of the aggregate type** | | | | | |
| Pleckstrin homology domain-containing family A member 2 | Plekha2 | 6.57±2.79 | *Ubiquilin-1* | *Ubqln1* | 8.27±2.65 |
| Tryptophan—tRNA ligase, cytoplasmic | Wars | 4.44±2.09 | Myosin phosphatase Rho-interacting protein | Mprip | 5.65±1.96 |
| DNA replication licensing factor MCM3 | Mcm3 | 3.73±1.22 | SAP domain-containing ribonucleoprotein | Sarnp | 5.56±1.78 |
| Eukaryotic translation initiation factor 2 subunit 1 | Eif2s1 | 3.65±0.71 | *Hsc70-interacting protein* | *St13* | 4.78±0.36 |
| E3 ubiquitin-protein ligase RNF126 | Rnf126 | 2.98±1.30 | Histone H3.1 | Hist1h3a | 4.30±1.28 |
| UBX domain-containing protein 1 | Ubxn1 | 2.66±1.21 | Clathrin interactor 1 | Clint1 | 4.28±0.91 |
| Proteasome subunit beta type-4 | Psmb4 | 2.60±0.63 | Coiled-coil-helix-coiled-coil-helix domain-containing protein 2 | Chchd2 | 4.02±0.73 |
| **Sequestosome-1** | **Sqstm1** | **2.47±0.03** | **RNA-binding protein FUS** | **Fus** | **3.26±1.23** |
| Interferon-inducible double-stranded RNA-dependent protein kinase activator A | Prkra | 2.40±0.75 | Tight junction protein ZO-1 | Tjp1 | 3.25±0.93 |
| Sorting nexin-3 | Snx3 | 2.36±0.29 | Ubiquitin-associated protein 2 | Ubap2 | 3.24±0.36 |
| Nuclear migration protein nudC | Nudc | 2.17±0.57 | *Small glutamine-rich tetratricopeptide repeat-containing protein alpha* | *Sgta* | 3.22±0.80 |
| Receptor of activated protein C kinase 1 | Rack1 | 2.17±0.28 | *DnaJ homolog subfamily B member 1* | *Dnajb1* | 3.11±1.11 |
| 40S ribosomal protein S2 | Rps2 | 2.12±0.54 | Chromobox protein homolog 1 | Cbx1 | 3.10±0.36 |
| Nuclear fragile X mental retardation-interacting protein 2 | Nufip2 | 2.11±0.51 | *Ubiquilin-2* | *Ubqln2* | **3.06±0.77** |
| 26S proteasome non-ATPase regulatory subunit 12 | Psmd12 | 2.09±0.67 | ***Phosphatidylinositol-binding clathrin assembly protein*** | ***Picalm*** | **2.81±0.25** |
| Vigilin | Hdlbp | 2.07±0.12 | CUGBP Elav-like family member 1 | Celf1 | 2.51±0.37 |
| Insulin-like growth factor 2 mRNA-binding protein 3 | Igf2bp3 | 2.06±0.78 | Transgelin-2 | Tagln2 | 2.17±0.49 |
| GTP cyclohydrolase 1 | Gch1 | 2.04±0.64 | RNA-binding protein 25 | Rbm25 | 2.16±0.51 |
| Fructose-bisphosphate aldolase A | Aldoa | 2.00±0.82 | Nucleolysin TIAR | Tial1 | 2.06±0.19 |
| 60S ribosomal protein L10 | Rpl10 | 1.99±0.40 | Caprin-1 | Caprin1 | 2.04±0.43 |
| ATPase WRNIP1 | Wrnip1 | 1.98±0.73 | Probable ATP-dependent RNA helicase DDX17 | Ddx17 | 2.02±0.18 |
| Ubiquitin fusion degradation protein 1 homolog | Ufd1l | 1.95±0.63 | Protein PRRC2C | Prrc2c | 2.01±0.48 |
| Proteasome subunit alpha type-6 | Psma6 | 1.93±0.64 | Ankyrin repeat domain-containing protein 17 | Ankrd17 | 1.95±0.26 |
| 40S ribosomal protein S27 | Rps27 | 1.91±0.34 | Pre-mRNA-processing factor 40 homolog A | Prpf40a | 1.82±0.23 |
| 26S proteasome non-ATPase regulatory subunit 3 | Psmd3 | 1.91±0.71 | DnaJ homolog subfamily C member 9 | Dnajc9 | 1.78±0.32 |
| Proteasome subunit beta type-7 | Psmb7 | 1.91±0.80 | Protein DEK | Dek | 1.78±0.76 |
| Adenine phosphoribosyltransferase | Aprt | 1.84±0.63 | *Hepatocyte growth factor-regulated tyrosine kinase substrate* | *Hgs* | 1.73±0.25 |

*(Continued)*

**Table 1.** (Continued)

| Enriched in polyGA$_{101}$ | | | Enriched in Httex1Q$_{97}$ | | |
|---|---|---|---|---|---|
| *Description* | *Gene ID* | *log2 enrichment (mean ± SD)* | *Description* | *Gene ID* | *log2 enrichment (mean ± SD)* |
| Cytochrome c oxidase subunit NDUFA4 | Ndufa4 | 1.82±0.28 | Ubiquitin-associated protein 2-like | Ubap2l | 1.65±0.20 |
| Interferon-induced protein with tetratricopeptide repeats 1 | Ifit1 | 1.76±0.50 | Nuclear pore complex protein Nup214 | Nup214 | 1.65±0.17 |
| Proteasome subunit beta type-5 | Psmb5 | 1.74±0.37 | Poly [ADP-ribose] polymerase 1 | Parp1 | 1.60±0.47 |
| 60S ribosomal protein L23 | Rpl23 | 1.73±0.20 | Calponin-3 | Cnn3 | 1.60±0.32 |
| T-complex protein 1 subunit eta | Cct7 | 1.72±0.38 | *DnaJ homolog subfamily A member 2* | *Dnaja2* | 1.58±0.63 |
| E3 ubiquitin-protein ligase TRIM32 | Trim32 | 1.72±0.40 | Serine/arginine repetitive matrix protein 2 | Srrm2 | 1.55±0.60 |
| ZW10 interactor | Zwint | 1.68±0.31 | Muscleblind-like protein 2 | Mbnl2 | 1.54±0.34 |
| Cyclin-dependent kinase 1 | Cdk1 | 1.59±0.30 | Protein phosphatase 1 regulatory subunit 12A | Ppp1r12a | 1.50±0.44 |
| ATP-dependent 6-phosphofructokinase, platelet type | Pfkp | 1.57±0.33 | *Poly(rC)-binding protein 1* | *Pcbp1* | 1.41±0.33 |
| Nuclear protein localization protein 4 homolog | Nploc4 | 1.54±0.61 | **TAR DNA-binding protein 43** | **Tardbp** | **1.41±0.15** |
| *Large proline-rich protein BAG6* | *Bag6* | 1.48±0.33 | Hexokinase-1 | Hk1 | 1.41±0.57 |
| 26S proteasome non-ATPase regulatory subunit 14 | Psmd14 | 1.48±0.30 | Poly(rC)-binding protein 3 | Pcbp3 | 1.35±0.16 |
| Malate dehydrogenase, cytoplasmic | Mdh1 | 1.48±0.22 | 5'-3' exoribonuclease 2 | Xrn2 | 1.32±0.36 |
| ADP-sugar pyrophosphatase | Nudt5 | 1.47±0.11 | Heterogeneous nuclear ribonucleoprotein F | Hnrnpf | 1.29±0.43 |
| 26S protease regulatory subunit 6A | Psmc3 | 1.42±0.44 | Pumilio homolog 1 | Pum1 | 1.22±0.07 |
| Bifunctional glutamate/proline—tRNA ligase | Eprs | 1.39±0.42 | Lamina-associated polypeptide 2, isoforms alpha/zeta | Tmpo | 1.22±0.46 |
| Aminoacyl tRNA synthase complex-interacting multifunctional protein 1 | Aimp1 | 1.30±0.49 | Tropomodulin-3 | Tmod3 | 1.21±0.37 |
| Ribosome-binding protein 1 | Rrbp1 | 1.26±0.39 | 14-3-3 protein beta/alpha | Ywhab | 1.15±0.43 |
| 40S ribosomal protein S27-like | Rps27l | 1.23±0.42 | Plectin | Plec | 0.97±0.20 |
| Ras GTPase-activating protein-binding protein 1 | G3bp1 | 1.18±0.39 | Small ubiquitin-related modifier 1 | Sumo1 | 0.83±0.22 |
| Glyceraldehyde-3-phosphate dehydrogenase | Gapdh | 1.15±0.29 | Regulator of nonsense transcripts 1 | Upf1 | 0.82±0.21 |
| Dihydropyrimidinase-related protein 2 | Dpysl2 | 1.09±0.32 | Vimentin | Vim | 0.72±0.13 |
| **Dihydropyrimidinase-related protein 3** | **Dpysl3** | **1.05±0.05** | Nuclear pore complex protein Nup98-Nup96 | Nup98 | 0.68±0.15 |
| Ataxin-10 | Atxn10 | 1.05±0.20 | Importin subunit alpha-1 | Kpna2 | 0.58±0.10 |
| ATP-binding cassette sub-family E member 1 | Abce1 | 1.04±0.25 | | | |
| 60S ribosomal protein L38 | Rpl38 | 1.03±0.29 | | | |
| Multifunctional protein ADE2 | Paics | 0.98±0.25 | | | |
| Polymerase delta-interacting protein 3 | Poldip3 | 0.97±0.11 | | | |
| Melanoma-associated antigen D1 | Maged1 | 0.94±0.23 | | | |
| Dihydropyrimidinase-related protein 1 | Crmp1 | 0.92±0.15 | | | |
| 60S acidic ribosomal protein P0 | Rplp0 | 0.89±0.28 | | | |
| ADP/ATP translocase 2 | Slc25a5 | 0.89±0.13 | | | |
| T-complex protein 1 subunit beta | Cct2 | 0.88±0.18 | | | |
| Eukaryotic translation initiation factor 3 subunit D | Eif3d | 0.87±0.26 | | | |
| T-complex protein 1 subunit delta | Cct4 | 0.84±0.22 | | | |
| Tubulin beta-5 chain | Tubb5 | 0.79±0.12 | | | |
| 26S protease regulatory subunit 10B | Psmc6 | 0.76±0.13 | | | |
| Golgi-associated plant pathogenesis-related protein 1 | Glipr2 | 0.75±0.19 | | | |
| IgE-binding protein | Iap | 0.75±0.19 | | | |
| Cell division control protein 42 homolog | Cdc42 | 0.71±0.13 | | | |

*(Continued)*

**Table 1.** (Continued)

| Enriched in polyGA$_{101}$ | | | Enriched in Httex1Q$_{97}$ | | |
| --- | --- | --- | --- | --- | --- |
| *Description* | *Gene ID* | *log2 enrichment (mean ± SD)* | *Description* | *Gene ID* | *log2 enrichment (mean ± SD)* |
| Non-POU domain-containing octamer-binding protein | Nono | 0.67±0.09 | | | |
| Aspartate aminotransferase, cytoplasmic | Got1 | 0.45±0.05 | | | |

* Only proteins that meet significance cut-off (hyperbolic curves, permutation-based FDR≤0.05, S0 = 0.1). Full table of proteins are shown in S2 Table; **Bold** are genes with known causes or risk factors for FTD-ALS (or other neurodegenerative diseases in the case of Picalm); *Italics* are cellular proteins previously seen to become more insoluble when Httex1Q$_{97}$ formed inclusions [73].

This result might be anticipated if these proteins partition into the aggregates based on their physicochemical properties rather than through a biological mechanism targeting inclusion assembly or clearance. Two of the proteins found exclusively in the polyGA aggregates (Phlda1 and Etfa) have been previously shown to have altered expression patterns in ALS models which may indicate a biological consequence of their coaggregation into the inclusions. Phlda1 was previously shown as upregulated in Fus-mutant motor neurons and suggested to operate as adaptive response to protect against apoptosis [42]. Phlda1 was also observed upregulated in sporadic ALS fibroblasts treated to stress compared to controls [43]. Etfa is a mitochondrial protein that was upregulated pre-symptomatically in a mouse transgenic SOD1 model of ALS [44] and downregulated after symptom development [45]. Hence the changed expression of Phlda1 and Etfa may arise from depleted activity arising from their sequestration into inclusions.

More notable connections to ALS biology were observed in the ensemble of proteins found in both the polyGA and Httex1 inclusions. Namely in *C9ORF72* mediated FTD-ALS brain tissue, a subset of inclusions is non-reactive to TDP-43 [46]. In most of the other forms of FTD-ALS that are not caused by *C9ORF72* mutation, TDP-43-reactive inclusions are a key pathological signature of neurons in disease [47]. TDP-43 negative inclusions were previously found to be immunoreactive for polyGA, suggesting they form by polyGA aggregation [48, 49]. We observed TDP43 preferentially enriched in the Httex1Q$_{97}$ inclusions raising the possibility that while TDP43 co-recruits to inclusions, it prefers associating with inclusions formed by proteins other than polyGA (Table 1). In addition, the TDP43-negative inclusions seen in vivo are immunoreactive to p62 [50] and lack immunoreactivity to FUS, optineurin, alpha-internexin and neurofilament [51, 52]. In our data p62 (also called sequestesome 1) is one of the most enriched proteins in polyGA inclusions, which is consistent with previous findings showing p62 binding to polyGA [10]. On the other hand Fus appeared diminished from polyGA inclusions by virtue of its enrichment in Httex1Q$_{97}$ inclusions, which has been observed previously in cell models of polyQ aggregation and human pathology [31, 53–55]. Hence these data point to the cell model of polyGA inclusions mimicking the process of aggregation and recruitment seen *in vivo* and also providing specificity of co-recruitment relative to Httex1Q$_{97}$.

Analysis of the differences is shown visually in Fig 2B by a (STRING) protein-protein interaction map and annotation to selected functional networks that cluster well with the STRING networks. Analysis of the genes enriched to either inclusion type yielded an enrichment for gene ontology and KEGG networks of microtubule cytoskeleton, proteasome complex, chaperones, RNA splicing and nuclear envelope (Fig 2B and S3 Table). The most significantly ranked GO terms show a notable cluster of protein quality control-related terms (Fig 2C and

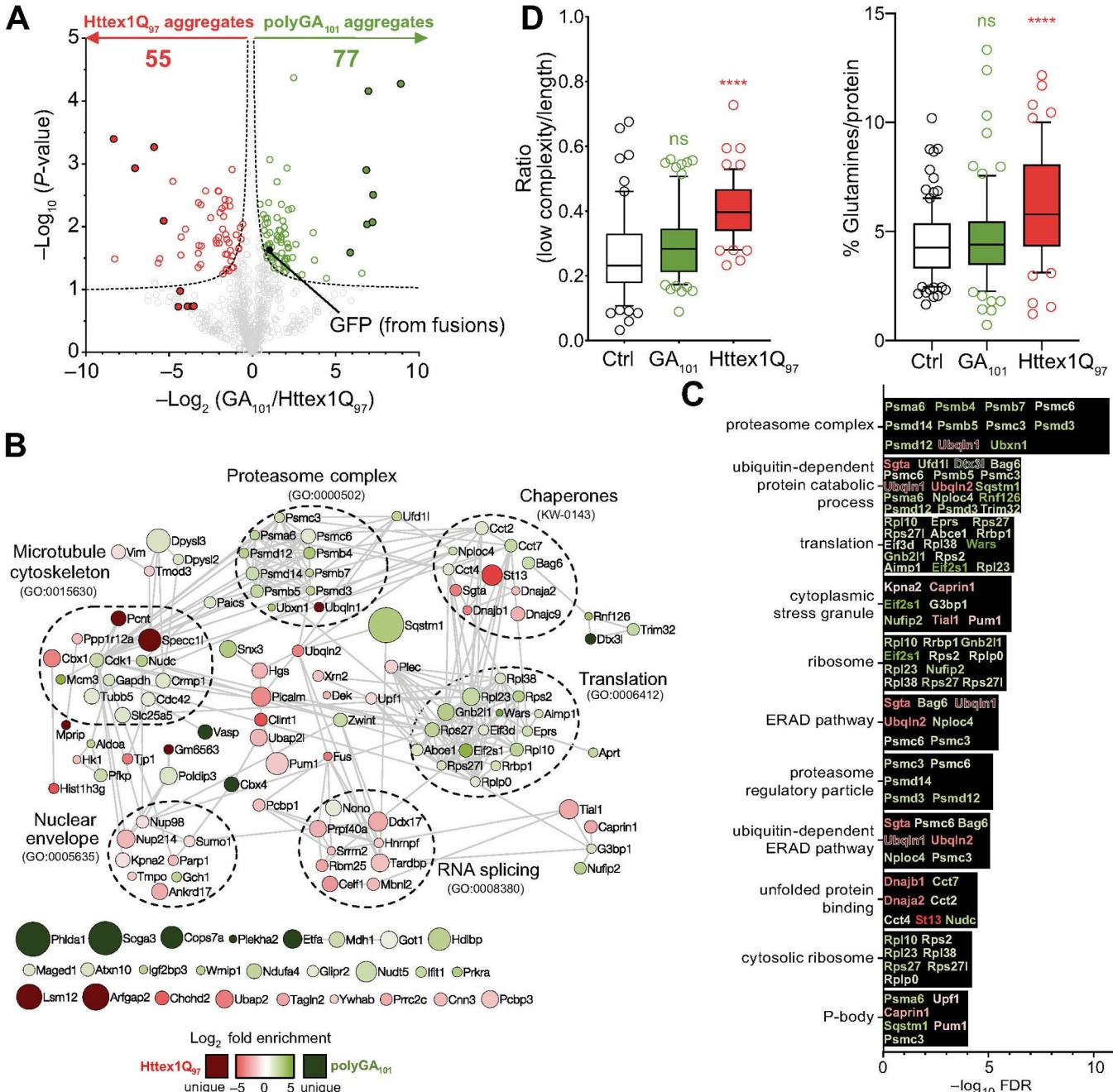

**Fig 2. Proteome recruitment patterns to polyGA_{101} and Httex1Q_{97} inclusions. A.** Volcano plot of proteins identified in the inclusions. *P*-values were calculated by a two-sided one samples t-test with null hypothesis that abundances were unchanged and the log_2 ratio was equal to 0. Proteins meeting stringency thresholds (hyperbolic curves, FDR≤0.05, S0 = 0.1) are shown as colored empty circles and proteins unique to each aggregate type are shown as filled colored circles. **B.** STRING interaction maps (v.11) determined in Cytoscape (v3.7) for proteins significantly enriched in the inclusions (the full list of proteins is in S2 Table). The analysis was done at the highest confidence setting (0.9). Each protein was represented by a colored circle sized proportionally to–log_{10} (*P*-value). The color scale represents logarithm of fold change. Selected significantly enriched GO terms (GOCC, GOPB, and UniProt keywords) are displayed (Full terms are shown in S3 Table). Note the proteins shown without connections at the bottom are those that are seen in the dataset but which do not have known protein interactions with the other proteins shown. **C**. The 10 most significantly enriched GO terms (from S3 Table) with proteins identified in the GO terms colour coded to enrichment (as per panel 2B). **D**. Analysis of enriched proteomes for low-complexity regions (IUPred-L) and high glutamine content. Significance of difference was assessed against a control dataset of random mouse proteins (S1 Table) with the Mann-Whitney-Wilcoxon test. Whiskers extend from 10 to 90%.

S3 Table). PolyGA was enriched with proteasome, ribosome and translation machinery suggesting that it might co-aggregate with newly synthesized proteins. Of 11 key GO terms shown in Fig 2C, 7 contained mixed enrichment patterns for proteins in Httex1 and polyGA inclusions suggesting that both aggregation types converge on mechanisms related to protein folding, quality control and degradation (GO terms of proteasome, cytoplasmic stress granule, ubiquitin-dependent protein catabolic process, unfolded protein binding, ERAD pathway, P-body, and ubiquitin-dependent ERAD pathway). These findings are in accordance with prior findings that protein aggregation impacts these biological processes and in particular an involvement in machinery for their clearance and degradation [56–59].

In addition, the data points more directly to proteins and genes implicated in FTD-ALS phenotype and mechanisms. Nudt5 was also found mildly enriched in polyGA inclusions (Table 1) and expression of this gene was significantly increased in motor neurons derived from induced pluripotent stem cells from ALS patients over controls [60]. Another protein of note mildly enriched in the polyGA aggregates was Dpysl3. A missense mutation that has been linked to ALS risk in the French population leads to shortened neuronal survival when expressed in cultured neurons in vitro [61]. Hence it remains plausible that co-aggregation of these proteins into polyGA inclusions sequesters their activity and renders cells less resilient to stress triggers.

The Httex1Q$_{97}$-enriched proteome also yielded noteworthy findings. Previously it was found that polyQ can preferentially co-recruit proteins containing intrinsically disordered domains (IDRs) and proteins enriched in glutamine [31, 54]. These patterns were also observed in our data (Fig 2D). However, polyGA did not show these enrichment patterns, indicative of specificity for polyQ in recruiting IDRs and Q-rich proteins. These data are consistent with polyQ more selectively co-aggregating other proteins enriched with glutamine. One such candidate is CREB Binding Protein, which is dysfunctional in Huntington's Disease. CREB Binding protein contains a glutamine repeat and is co-aggregated into inclusions formed by mutant Htt [62–66]. Hence the co-aggregation of CREB binding protein may result in its loss of activity as one contribution to pathogenesis.

To assess whether the changes in polyGA inclusion formation had other effects on proteome abundance, we expressed polyGA$_{101}$ and at 48 h after transfection sorted live cells into those with visible aggregates from those without by a flow cytometry sorting method called pulse shape analysis [67] (Fig 3A). Pulse shape analysis uses the fluorescence signal width and height parameters to infer whether the protein has a diffuse localization (ni) or condensed localization when inclusions form (i). GFP alone does not aggregate and established the reference (ni) gate for when inclusions form (Fig 3A). PolyGA-GFP fusions formed both i and ni populations, which we attribute as cells with inclusions and those without respectively (Fig 3A). Assessment of cells with the dye Sytox, which selectively labels the nuclei of dead and dying cells, revealed cells with inclusions (the i gate) were more reactive to Sytox than cells with soluble polyGA (ni gate) (Fig 3A inset). We hence excluded cells that were dying from analysis. As a result, 35% of the remaining live cells expressing polyGA had inclusions (Fig 3A). We sorted these live cells into those with polyGA inclusions versus cells with soluble polyGA at matched median expression levels (red population in right panel of Fig 3A) for proteomic analysis. Fig 3B shows the proteomic abundances for cells matched for total harvested protein levels (i.e. polyGA inclusions versus soluble polyGA). Out of 2420 proteins identified, we observed 56 proteins that significantly changed abundance in these sorted cell populations (Fig 3B and S4 Table). There was an enrichment of GFP peptides (derived from the GFP-polyGA$_{101}$ fusion) in the cells with inclusions, which could arise if the inclusions lead to the GFP becoming quenched. Hence, we caution that the difference in GFP abundance may contribute to some of the conclusions that can be drawn from this data. However, there was

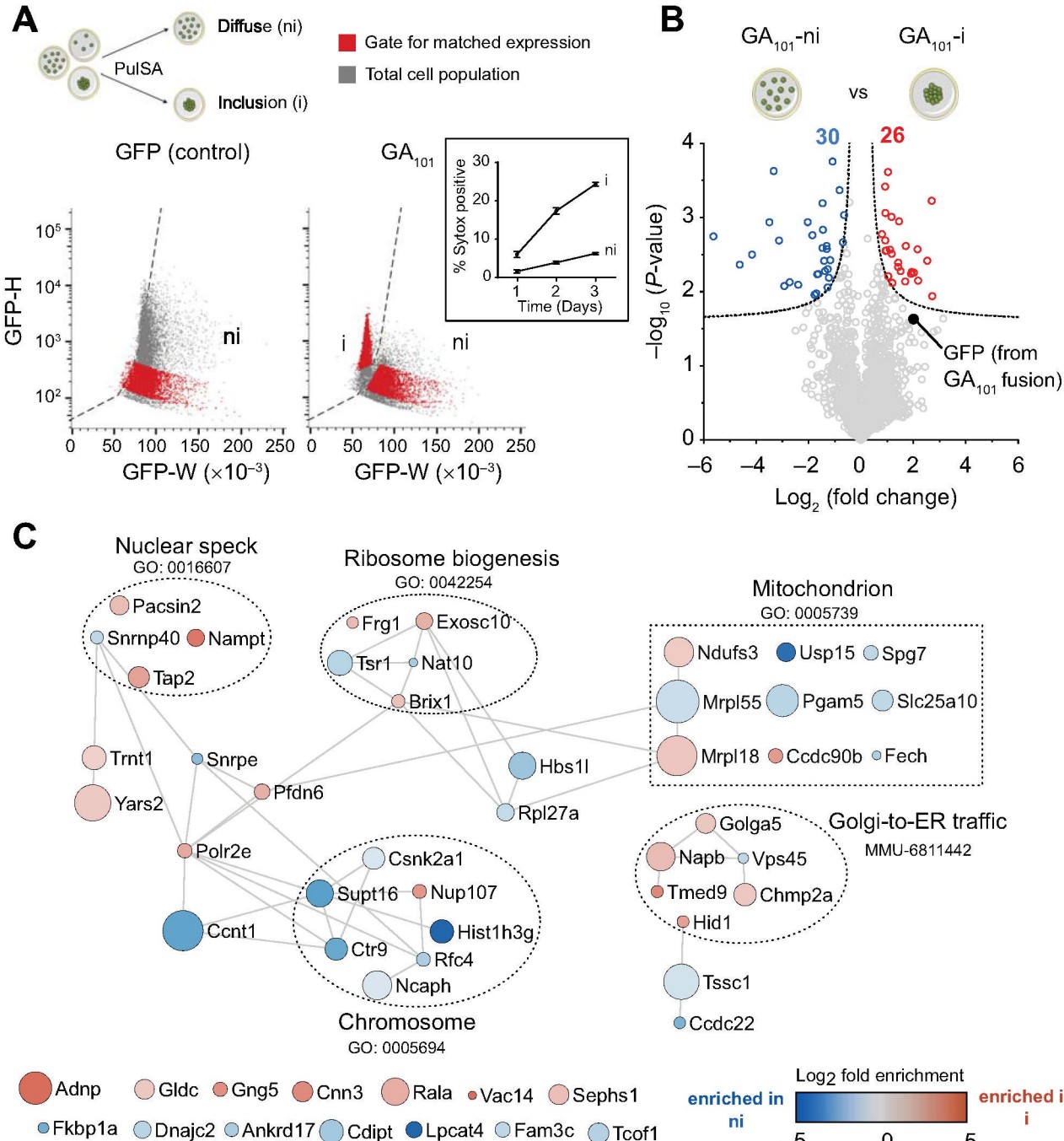

**Fig 3. Cellular protein abundance changes arising from polyGA$_{101}$ aggregation. A.** Schematic of flow cytometry method of pulse shape analysis (PulSA) to sort cells enriched with inclusions (i) from those without inclusions (ni). Cells with inclusions display shorter width (W) fluorescence values versus cells with soluble protein, and typically higher height values (H) arising from the condensed foci of fluorescence inside the cells. Cells were sorted to exclude dead cells by DAPI reactivity (which–like Sytox–labels the nuclei of dead and dying cells). Inset shows percentage of transfected cells (for polyGA$_{101}$-GFP) reactive to Sytox by time after transfection. $n = 4$, means ± SD shown. **B.** Volcano plots of proteins that changed their abundance upon polyGA aggregation. Shown are cells without polyGA$_{101}$ aggregates (i population) versus cells with soluble polyGA$_{101}$ (ni population) collected at matched median GFP fluorescence. The dotted line indicates significance cut-off (hyperbolic curves, FDR≤0.05, S0 = 0.1) and proteins meeting stringency thresholds are shown as colored circles. **C.** Protein-protein interaction network (STRING v11) of proteins significantly changed in abundance upon polyGA aggregation (i.e. polyGA$_{101}$ aggregates (i population) versus cells with soluble polyGA$_{101}$ (ni population)). The full list of proteins are in S4 Table. The analysis was done at the highest confidence setting. Each protein was represented by a colored circle sized proportionally to -log$_{10}$ (*P*-value). The color scale represents logarithm of fold change. Selected significantly enriched GO terms (GOCC, GOPB, and UniProt keywords) are displayed (S5 Table).

no overlap in the proteins seen enriched in polyGA inclusions (data in Fig 2 and S3 Table) with proteins that changed abundance when polyGA shifted from a soluble to aggregated state (Fig 3C and S4 Table). This provides firmer confidence that the enrichment seen in the polyGA aggregates arises from co-aggregation rather than changes in gene expression.

Of the genes that changed expression, protein interaction networks yielded significant enrichment in networks including nuclear speck (GO: 0016607), ribosome biogenesis (GO: 0042254), chromosome (GO:0005694), mitochondrion (GO:0005739) and Golgi-to-ER-traffic (MMU-6811442) (S5 Table). Some of these pathways would be anticipated to be correlated to stress responses incurred by protein aggregation or proteostasis imbalance based on links between ribosome biogenesis and nucleolar stress response [68], ribosome biogenesis and proteostasis imbalance [69], mitochondria as mediators of apoptosis [70], and role of the Golgi as a stress sensor in neurodegeneration [71]. However, we did not note any striking changes that pertained to novel mechanisms other than that from this data.

Lastly, we investigated the overlap of proteins enriched in Httex1Q$_{97}$ inclusions with proteins identified in Httex1 inclusions from other studies. Mitsui et al [72] identified 8 prominent proteins in purified Httex1Q97 inclusions that eluted by SDS-PAGE (HSP84, HSC70, α-tubulin, β-tubulin, EF-1α, HDJ-1 and HDJ-2 and actin). We observed all of these in our study suggesting they are enriched in both polyGA and Httex1 inclusions. HDJ-1 and HDJ-2 are Hsp40 family protein members and we observed significant enrichment of Hsp40 proteins dnaja2, dnajb1 and dnajc9 in the Httex1Q$_{97}$ inclusions (S2 Table). Of the others identified by Mitsui et al we saw no specific enrichment to polyQ inclusions, which suggests that these proteins are recruited to both inclusion types. This conclusion is supported by the enrichment of HSP84 (which is Hsp90a) and EF1α (which is Eef1a) immunoreactivity to the surface of the inclusions [72].

We also examined the overlap with our previously reported changes in solubility of whole cell proteome before versus after inclusions had formed [73] (Fig 4A). In that dataset (Sui et al [73]) we observed 25 proteins that significantly decreased in solubility as cells expressing Httex1Q$_{97}$ shifted from a dispersed unaggregated state to forming inclusions [73] (S6 Table). Of these, 9 proteins were found in our list of 55 proteins significantly enriched in Httex1Q$_{97}$ inclusions (Pcbp1, Dnaja2, Sgta, Picalm, Hgs, Clint1, Ubqln1, Ubqln2 and Dnajb1) (Fig 4A). When we also considered proteins that were identified in either inclusion (full list of proteins in S2 Table), which are therefore candidate proteins that are recruited to both inclusion types, we found a further 7 proteins that overlapped with the previous published data from Sui et al [73]. Analysis of the protein:protein interaction networks by STRING analysis revealed two robust networks within these proteins that map onto gene ontology enrichments for mechanisms related to protein quality control including positive regulation of proteolysis (GO:0045862; FDR of 0.0029), positive regulation of ERAD pathway (GO:1904294; FDR of 8.76E-05), heat shock protein binding (GO:0031072; FDR of 2.43E-06) and protein folding (GO:0006457; FDR of 0.00034) (Fig 4B and full list of GO terms in S6 Table).

Clint1 and Ubqln2 were previously shown to colocalize to polyQ inclusions, supporting this conclusion [54, 74]. An interesting note with respect to mechanism is that UBQLN2 targets ubiquitinated substrates for degradation in ERAD and autophagy [75]. Furthermore mutations in UBQLN2 cause ALS, which appear to lead to an impairment in the degradation of ubiquitinated proteins [76]. Further supporting an important role linking protein aggregation and degradation more broadly to these neurodegenerative diseases is the enrichment of Picalm in the polyQ inclusions. Picalm is a phosphatidylinositol-binding clathrin assembly protein and has been shown from GWAS to be a top ten risk for Alzheimer's disease [77, 78]. It has been reported to modulate intracellular APP processing and plaque pathogenesis [79], modulate autophagy and alter tau clearance [80].

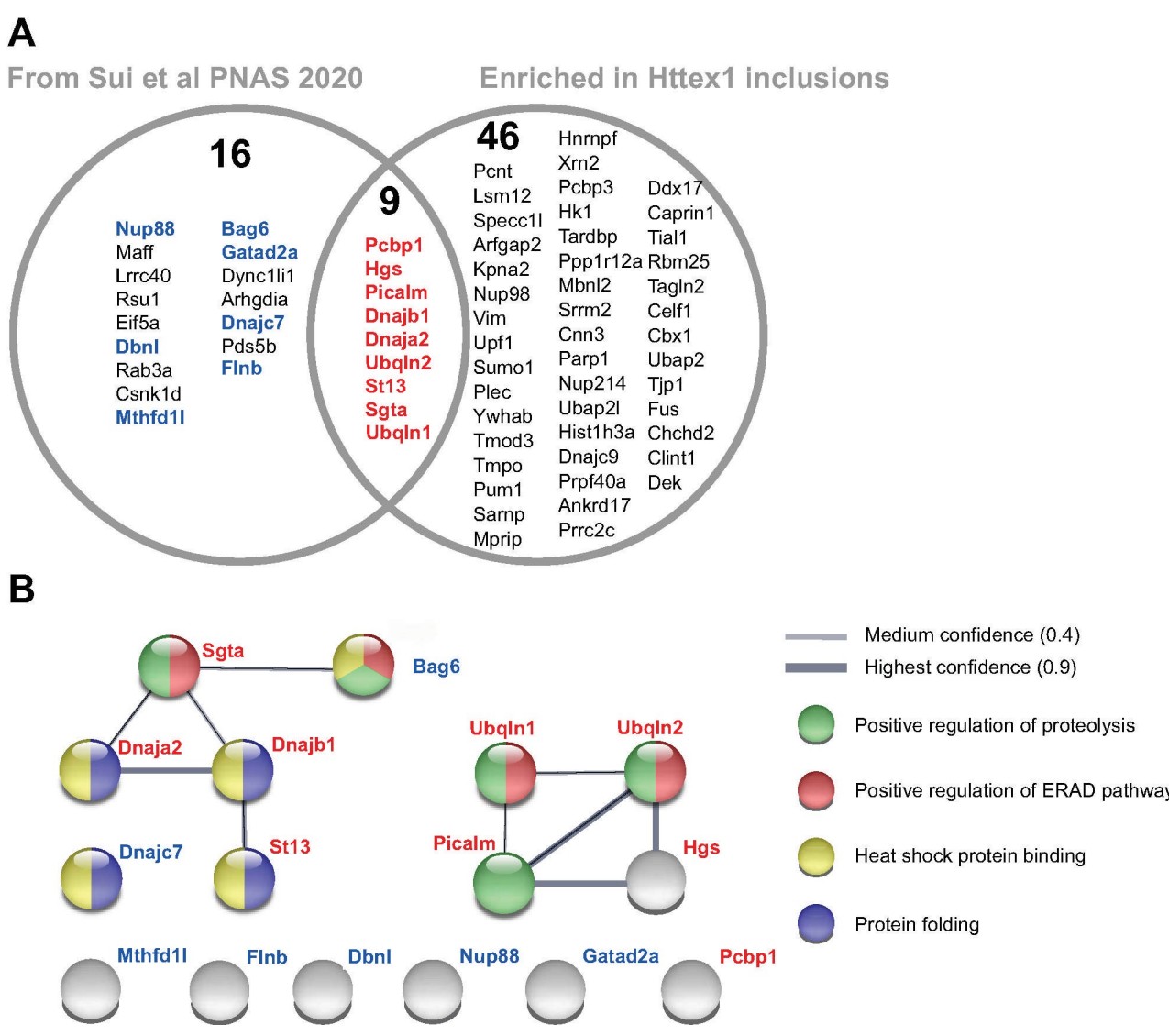

**Fig 4. Commonalities of proteins found in Httex1Q$_{97}$ and polyGA$_{101}$ inclusions with changes in proteome solubility due to Httex1 aggregation.** **A**. Venn diagram of proteins previously found by Sui et al [73] to change solubility when Httex1 formed inclusions compared to proteins enriched in Httex1 inclusions over polyGA inclusions (our data here in S2 Table). Proteins highlighted in bold blue are additional proteins seen in our dataset that are not enriched to Httex1. **B**. STRING (v11.0) interaction network of the proteins from panel A. Protein names are colour coded to match panel A. Shown are interactions at highest (0.9) and medium confidence settings (0.4). Note the proteins shown without connections are those that are seen in the dataset but which do not have known protein interactions with the other proteins shown. Protein nodes are colour coded to selected significant Gene ontology terms. Full list of GO terms for Biological Process and Molecular Function are shown in S6 Table.

Collectively the data here reports proteins that co-aggregate into two very different neurodegenerative disease proteinaceous deposits. The findings provide specificity of proteins to the aggregation type that provide useful perspective to that reported by others. Moreover, the mechanisms of protein clearance and regulation of protein folding-misfolding appear relevant to both aggregation types and notably of a number of proteins in the Httex1Q$_{97}$ aggregates that when mutated are modifiers of ALS risk [81] (list of ALS genes, and mouse protein counterparts, are shown in S7 Table). Therefore, the findings identify a synergy of biological mechanisms involved in protein folding quality control and degradation that appear central to at least two different neurodegenerative diseases, and possibly more applicable to the other neurodegenerative diseases involving inappropriate protein aggregation.

## Supporting information

**S1 Table. List of random proteins from mouse Uniprot database.**
(XLSX)

**S2 Table. Proteins enriched in inclusions of polyGA$_{101}$ and Httex1Q$_{97}$.** Relates to Table 1 and Fig 2.
(XLSX)

**S3 Table. Gene ontology terms enriched among proteins identified in polyGA$_{101}$ or Httex1Q$_{97}$ inclusions.** Relates to Fig 2.
(XLSX)

**S4 Table. Cellular abundances of proteins caused by polyGA$_{101}$ aggregation.** Relates to Fig 3.
(XLSX)

**S5 Table. Gene ontology terms enriched among proteins that changed abundance upon polyGA$_{101}$ aggregation.** Relates to Fig 3.
(XLSX)

**S6 Table. Gene ontology terms of enrichment for commonalities of proteins found in Httex1Q$_{97}$ and polyGA$_{101}$ inclusions with changes in proteome solubility due to Httex1 aggregation.** Relates to Fig 4.
(XLSX)

**S7 Table. List of human ALS genes and mouse protein counterparts derived from Table 1 in Nguyen et al Trends Genet.** 34(6) (2018) 404–423.
(XLSX)

## Acknowledgments

We thank the Bio21 Melbourne Mass Spectrometry and Proteomics facility for their technical assistance.

## Author Contributions

**Conceptualization:** Danny M. Hatters.

**Data curation:** Mona Radwan, Jordan D. Lilley.

**Formal analysis:** Mona Radwan, Ching-Seng Ang, Gavin E. Reid, Danny M. Hatters.

**Investigation:** Mona Radwan, Jordan D. Lilley.

**Methodology:** Mona Radwan, Ching-Seng Ang, Gavin E. Reid.

**Project administration:** Danny M. Hatters.

**Resources:** Ching-Seng Ang, Danny M. Hatters.

**Supervision:** Gavin E. Reid, Danny M. Hatters.

**Visualization:** Mona Radwan.

**Writing – original draft:** Mona Radwan, Danny M. Hatters.

**Writing – review & editing:** Ching-Seng Ang, Gavin E. Reid, Danny M. Hatters.

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
