## [Decision Letter · Decision Letter 0]

5 Jun 2020

PONE-D-20-12595

Inclusion bodies formed by polyglutamine and poly(glycine-alanine) are enriched with distinct proteomes but converge in proteins that are risk factors for disease and involved in protein degradation

PLOS ONE

Dear Dr. Hatters,

Thank you for submitting your manuscript to PLOS ONE. After careful consideration, we feel that it has merit but does not fully meet PLOS ONE’s publication criteria as it currently stands. Therefore, we invite you to submit a revised version of the manuscript that addresses the points raised during the review process.

Three expert reviewers have carefully examined the manuscript, and have provided detailed critiques and suggestions. They express enthusiasm about the experimental data, but also identify a number of points in the manuscript that need clarification. As such, the manuscript in its current state does not fulfil the publication criteria of PLOS ONE. Primarily, criteria 3 and 4 (description of the experiments and analysis; support of the written conclusions) are not met.

Please provide a point-by-point response to all the comments identified by the authors. One key question relates to the comments from reviewer 3, who notes that the different experiments were performed at distinct time points, which may affect the ability to integrate the respective findings. The reviewers also comment on a need to clarify or rephrase a number of claims made, including in the abstract.  In particular, there are some questions about whether the reasoning or evidence behind certain mechanistic claims are completely clear. Among other significant questions, I also note some issues where the text appears to be unclear about which conditions or samples are being compared in comparative statements in the paper (see reviewer comments).

We look forward to receiving your revised manuscript.

Kind regards,

Patrick van der Wel, PhD

Academic Editor

PLOS ONE

Journal Requirements:

Additional Editor Comments (if provided):

Reviewers' comments:

Reviewer's Responses to Questions

**Comments to the Author**

1. Is the manuscript technically sound, and do the data support the conclusions?

Reviewer #1: Yes

Reviewer #2: Yes

Reviewer #3: Yes

2. Has the statistical analysis been performed appropriately and rigorously? 

Reviewer #1: Yes

Reviewer #2: I Don't Know

Reviewer #3: Yes

3. Have the authors made all data underlying the findings in their manuscript fully available?

Reviewer #1: Yes

Reviewer #2: Yes

Reviewer #3: Yes

4. Is the manuscript presented in an intelligible fashion and written in standard English?

Reviewer #1: Yes

Reviewer #2: Yes

Reviewer #3: Yes

5. Review Comments to the Author

Reviewer #1: In the article titled, “Inclusion bodies formed by polyglutamine and poly(glycine-alanine) are enriched with distinct proteomes but converge in proteins that are risk factors for disease and

involved in protein degradation”, Radwan and coworkers use state-of-the-art proteomics techniques to evaluate the proteins that co-purify with specific proteinaceous aggregates. They focus on a polyglutamine-expanded fragment of Huntingtin protein exon 1 (Httex1Q97) and a glycine-alanine expansion (polyGA101), which respectively are models for Huntington’s disease and ALS (caused by expansions in C9orf72). These pathological proteins were expressed in cell culture and the authors isolated the aggregates for proteomic and bioinformatic analyses. The major observation is that these pathological proteins formed distinct aggregates within cells and each type of aggregate contained a unique set of enriched co-aggregating proteins. Thus, the aggregation of each species has the potential to interfere with cell machinery differently and cellular pathology may occur via protein-specific mechanisms. The authors also include an important control for polyGA101 in which they determine that co-aggregation of specific proteins is not a result of protein over-expression.

Overall, the work is clearly presented and provides new data sets for understanding potential pathological mechanisms associated with protein aggregation. The authors provide some speculation about the significance of certain co-aggregating proteins, but this analysis seems incremental in its advancement of knowledge. However, the authors do not over-interpret their data.

Questions for authors:

Were there any proteins that were identified only in HttexQ97 or polyGA101 samples? The Supplementary Table 2 suggests the authors identified 737 total proteins and all of these proteins were present in both samples. Perhaps the enriched proteins that were discussed in the text only make up a small percentage of the proteins that were co-aggregating in both types of samples? If that were the case, their significance would be diminished. It also seems possible that some proteins could be completely specific to only one type of aggregate? Making these points clearer in the text might make it easier for the reader to interpret.

Major suggestion:

A more quantitative presentation of the data in Figure 1B is needed. It is not clear how representative this image is of its larger cell population.

Minor suggestions:

On Line 243, the authors should emphasize that the proteins are enriched relative to the other type of aggregate (Htt vs polyGA). It’s possible to read this paragraph and think the proteins are enriched over “background”.

In the Figure 1A legend, specify the number of cells used for the graph and include the p-value.

On Line 276, I think the authors intend to reference Figure 2C.

Reviewer #2: This manuscript by Radwan and colleagues describes a well-designed set of experiments to determine the differences in the composition of polyQ and polyGA inclusions and to study the differences in the total proteome of cells with and without polyGA inclusions. It will provide a valuable resource for researches in the fields of both Huntington’s Disease and Frontotemporal dementia/amyotrophic lateral sclerosis. I recommend this manuscript for publication without further experiments, but I believe that a few points need to be explained/described in more detail:

It is not clear to me why the authors describe the disease caused by C9ORF72 GGGGCC hexanucleotide repeat expansion as motor neuron disease and not as Frontotemporal dementia and/or amyotrophic lateral sclerosis 1. This decision should be explained in the manuscript, and the connection between C9orf72 and FTD/ALS has to be mentioned prominently.

The abstract mentions that “both structures revealed a synergy of degradation machinery” and in the last paragraph it is stated that the “mechanisms of protein clearance mechanism appear relevant to both aggregation types”. It is not clear to me what is meant by that, and I have a hard time finding data addressing this point in the results section of the manuscript. It is shown that the proteasome is enriched in polyGA aggregates, but I can’t find any other degradation pathways regarding polyQ.

Previous studies on the composition of polyQ and polyGA inclusion bodies should be mentioned more prominently, including the work from N. Nukina that also used FACS sorting to isolate polyQ inclusions (Mitsui et al 2002).

While not absolutely necessary, I think it would increase the value of this study if the overlap and differences between the published datasets and these new datasets would be discussed.

The sentence in 274 needs clarification. “Overall both inclusions yielded an enrichment for gene ontology and KEGG networks of microtubule cytoskeleton, proteasome complex, chaperones, RNA splicing and nuclear envelope (Fig 2C; Table S3).” To me this sounds like both inclusions are enriched in these pathways whereas Figure 2 and Table 2C show GO terms enriched in either one kind of inclusion or the other.

GFP is among the enriched proteins in the polyGA interactome, and potentially 2 times more polyGA could also mean two times more background binders. The authors should explain in more detail how this can influence their analysis, and what steps have been taken to control for that. I don’t think that it is invalidating the analysis, but I believe it is important to discuss potential problems.

The authors argue that the Go terms listed in 3C and S5 indicate a stress response. Can they explain what they mean with this, and give some references where these terms are connected to stress response pathways.

Minor points:

- Sentence in line 41/42 is incomplete

- 276: The Figure described is not 2C.

- Sentence in line 345/346 is incomplete

Reviewer #3: In this current article, the authors evaluate proteinaceous inclusions known to be present in C9orf72-ALS and Huntington’s Diseases. Initially, they investigate the time of formation of C9orf72-ALS associated poly-GA inclusions and show that their rate of formation is slower than that of poly-Q aggregates found in Huntington’s. Following isolation of these aggregates, the authors then showed an impressive mass spectrometry analysis of proteins enriched in these two instances, along with STRING interaction maps showing involved cellular pathways. The authors also then provided a secondary mass spectrometry analysis, using the novel PulSA method of distinguishing between cells containing diffuse and aggregated poly-GA. The results from these various experiments will be valuable in identifying protein targets and pathways involved both in Huntington’s, C9orf72-ALS, and potentially a broader range of neurodegenerative diseases involving protein degradation.

Strengths:

The abstract and introduction are both concise and informative, detailing all key findings of the paper. The manuscript details appropriate inclusion of statistical methods, availability of data sets to the public, and justification of replicates and controls used. Figure 1 demonstrates an impressive ability to image and evaluate the same cells every 15 minutes for 4 days straight. Figures 2A and 3B both show distinct proteomal enrichments with striking volcano plots. Key proteins and pathways are identified associated within poly-GA and polyQ inclusions, which will be informative in guiding future studies and potential therapeutics for these diseases.

Major points of concern:

The use of PulSA is novel and allows for meaningful and insightful results in the given manuscript. However, it is unclear why there is a shift in timing from the 24hr collection of aggregates for “aggregate enriched” mass spec (starting line 91), and a later collection time of 48hr for PulSA analysis (starting line 107). There is no justification for this in the text. The highlighted takeaway from Figure 1 was that very few GA expressing cells has detectable GA aggregates at the 24hour timepoint, and that almost all had formed inclusions by 60hr. It is confusing therefore as to why the 24 timepoint was used for the staining of inclusions in Figure 1 part B, as well as the aggregates purifies for mass spec in Figure 2 when so few inclusions were present. The timepoint then shifted to a 48-hr timepoint for the GA-influence proteome mass spec evaluated in Figure 3, without a rationale or explanation.

More explicit description is required regarding the “whole proteome” alterations associated with GA aggregation for Figure 3B. On the basis of the in-text and figure legend description, it seems that the comparison that was performed is solely the proteome changes that occur between cells containing GA soluble versus GA insoluble inclusions. However, the volcano plot in Fig 3B seems to also compare the proteome outcomes of these two GA states to GFP alone. It is pivotal to understand which of these possibilities has been evaluated. Proteomic changes are likely to occur comparing GA soluble to GFP or control non-transduced cells, and so this must be explicitly addressed. Evaluation and consideration of the publicly available dataset related to Figure 3B needs to be done in light of the actual experimental setting performed.

Minor points of concern:

The introduction was thorough yet brief. Overall it seems that the proteome changes of poly-GA versus the arginine-containing DPRs should be highlighted a bit more. Furthermore, presenting a stronger case for this study comparing poly-GA versus poly-Q will be meaningful, given the very distinct pathologies of C9-ALS versus Huntington’s Disease, and that the overall proteome changes for both individual disease phenotypes have already been established in Neuro2a cells and primary neurons.

It is unclear as to why additional proteins are included to left of the STRING interaction maps in Figure 2B, this should be noted in the figure legend. Presumably this figure includes all 48 polyQ-enriched and all 69 poly-GA enriched proteins from portion A.

The prior known association of poly-GA dipeptides with SQSTM1/p62 should be noted either in line 270, or in the description of Figure 2B results (May Acta Neuropathol 2014).

The paragraph involving Httex1Q97 enriched proteome (starting line 290) could be expanded upon with implications.

The “in-text” description of the exclusion of cells for Figure 3 is unclear (line 299). It should be clarified within the main text, as was done in the figure legend (line 321), to allow for better understanding of which cells were actually evaluated.

Does the statement in lines 300-304 “A lower yield of aggregates than we measured by live cell imaging” imply that some cells recorded in the live imaging experiments were actually “almost dead” and would have been Sytox positive if tested?

The statement “there was no overlap in the proteins seen enriched in polyGA inclusions with proteins that changed expression due to polyGA aggregation” requires an explanation into how this comparison was performed.

The authors mention “lastly we investigated the overlap of proteins enriched in Httex1Q97 inclusions” (line 333). While there is a description of the results and highlighting of key findings, there is no figure representing this data.

The description of Picalm found in polyQ inclusions, and the link to Alzheimer’s disease is an interesting one (line 345-349). As the implication is a “broad link to neurodegenerative diseases” it would also be intriguing to know if Picalm was also found in the poly-GA inclusion protein list.

Conclusion: The statement starting on line 352, “the mechanisms of protein clearance” needs to be highlighted more specifically with examples. This seems to be one of the key conclusions that the authors wish to put forward, however it is not extremely well supported from the reading of the text at this time. Additionally, at the end of this statement it would be beneficial to include of list of the mutated proteins which are modifiers of MND risk (line 354). Overall implications and next steps also are lacking, as to how this data may be applied to future studies.

Overall, the paper would benefit from a thorough proofreading to rectify the following errors and potentially more that the authors will identify:

• Methods: NanESI-LC-MS analysis. Starting at line 156, it is unclear what “B” is. It seems that it may be 100% CH3CN/5% DMSO/0.1% formic acid v/v as listed. The end of this line also states “B” to “B” multiple times in describing the percentages of solutions used for an elution gradient. Should these say A to B, or are B different percentages of eluent B truly used?

• Line 276 mislabels Figure 2B as 2C

6. PLOS authors have the option to publish the peer review history of their article (what does this mean?). If published, this will include your full peer review and any attached files.

Reviewer #1: No

Reviewer #2: No

Reviewer #3: No

---

## [Author Response · Author response to Decision Letter 0]

9 Jul 2020

please see separate response to reviewers document

---

## [Decision Letter · Decision Letter 1]

11 Aug 2020

PONE-D-20-12595R1

Immiscible inclusion bodies formed by polyglutamine and poly(glycine-alanine) are enriched with distinct proteomes but converge in proteins that are risk factors for disease and involved in protein degradation

PLOS ONE

Dear Dr. Hatters,

Thank you for submitting your manuscript to PLOS ONE. After careful consideration, we feel that it has merit but does not fully meet PLOS ONE’s publication criteria as it currently stands. Therefore, we invite you to submit a revised version of the manuscript that addresses the points raised during the review process.

I thank you for the extensive revisions in the manuscript that address almost all of the prior points (based on my own evaluation and one of the prior reviewers who was available despite the summer and COVID constraints). Please find a few smaller concerns listed in the reviewer comments, which should be clarified or rectified prior to publication. I think most comments speak for themselves and are minor corrections. I do want to comment on the first issue: the use of the term "synergy". The response letter and revised text clarify that you see evidence of multiple quality control (QC) mechanisms being involved/activated in both HttEx1 and polyGA-affected cells. That much is clear.  However, I agree with the reviewer that the data do not (without further explication) provide obvious (or any?) evidence of a true "synergy" of these mechanisms. I.e. the data show that proteins from multiple QC mechanisms are present and relevant (although unable to resolve the inclusions). However, as far as I can tell the data do not show that their combined action is "better" than the sum of their individual activities (= the definition of synergy). Can you please clarify specifically the evidence of synergistic effects, or rephrase the use of term synergy? (E.g. one could consider the use of phrases like "a combined action" or "collaborative involvement" or similar, which would not imply true "synergistic" effects)

Otherwise, please note that the journal PLOS ONE does not offer further editorial services, such that this may present your last chance to correct grammatical and other errors as these will not be addressed in any final typesetting process provided by the journal. 

We look forward to receiving your revised manuscript.

Kind regards,

Patrick van der Wel, PhD

Academic Editor

PLOS ONE

Reviewers' comments:

Reviewer's Responses to Questions

**Comments to the Author**

1. If the authors have adequately addressed your comments raised in a previous round of review and you feel that this manuscript is now acceptable for publication, you may indicate that here to bypass the “Comments to the Author” section, enter your conflict of interest statement in the “Confidential to Editor” section, and submit your "Accept" recommendation.

Reviewer #2: (No Response)

2. Is the manuscript technically sound, and do the data support the conclusions?

Reviewer #2: Yes

3. Has the statistical analysis been performed appropriately and rigorously? 

Reviewer #2: I Don't Know

4. Have the authors made all data underlying the findings in their manuscript fully available?

Reviewer #2: Yes

5. Is the manuscript presented in an intelligible fashion and written in standard English?

Reviewer #2: Yes

6. Review Comments to the Author

Reviewer #2: Most of my comments have been addresses sufficiently. The only point where I am still unhappy is the synergy aspect mentioned in the abstract. Even after the revisions and the additional inclusion of the term protein folding, I fail to see how the results presented here demonstrate a synergetic effect. Perhaps it is possible to phrase this differently.

Minor points:

Line 19/20: the protein number given in the abstract for proteins associated with both inclusion types is identical to the number given in table S2 for proteins associated with both inclusion types plus the proteins identified for only one kind of inclusion.

line 55: I am not sure what mechanism is meant by the phrase “assemble by an amyloid-like mechanism”, and believe that the original phrase that the inclusions were “amyloid-like” better describes what is known about polyQ and polyGA inclusions.

line 338: please exchange [63,64,65-67] with [63-67]

lines 585/619: there are links present that are inconsistent with the formatting of the other parts of the reference section

7. PLOS authors have the option to publish the peer review history of their article (what does this mean?). If published, this will include your full peer review and any attached files.

Reviewer #2: No

---

## [Editor Report · Decision Letter 2]

14 Aug 2020

Immiscible inclusion bodies formed by polyglutamine and poly(glycine-alanine) are enriched with distinct proteomes but converge in proteins that are risk factors for disease and involved in protein degradation

PONE-D-20-12595R2

Dear Dr. Hatters,

We’re pleased to inform you that your manuscript has been judged scientifically suitable for publication and will be formally accepted for publication once it meets all outstanding technical requirements.

Kind regards,

Patrick van der Wel, PhD

Academic Editor

PLOS ONE
---

## [Editor Report · Acceptance letter]

18 Aug 2020

PONE-D-20-12595R2 

Immiscible inclusion bodies formed by polyglutamine and poly(glycine-alanine) are enriched with distinct proteomes but converge in proteins that are risk factors for disease and involved in protein degradation 

Dear Dr. Hatters:

I'm pleased to inform you that your manuscript has been deemed suitable for publication in PLOS ONE. Congratulations! Your manuscript is now with our production department. 

Kind regards, 

on behalf of

Dr. Patrick van der Wel 

Academic Editor

PLOS ONE